# A Neural Model for Compositional Word Embeddings and Sentence Processing

**Jean-Philippe Bernardy** and **Shalom Lappin**
Centre for Linguistic Theory and Studies in Probability
Department of Philosophy, Linguistics and Theory of Science
University of Gothenburg
jean-philippe.bernardy@gu.se      shalom.lappin@gu.se

## Abstract

We propose a new neural model for word embeddings, which uses Unitary Matrices as the primary device for encoding lexical information. It uses simple matrix multiplication to derive matrices for large units, yielding a sentence processing model that is strictly compositional, does not lose information over time steps, and is transparent, in the sense that word embeddings can be analysed regardless of context. This model does not employ activation functions, and so the network is fully accessible to analysis by the methods of linear algebra at each point in its operation on an input sequence. We test it in two NLP agreement tasks and obtain rule like perfect accuracy, with greater stability than current state-of-the-art systems. Our proposed model goes some way towards offering a class of computationally powerful deep learning systems that can be fully understood and compared to human cognitive processes for natural language learning and representation.

## 1   Introduction

The word embeddings that deep neural networks (DNNs) learn are encoded as vectors. The various dimensions of the vectors correspond to distributional properties of words, as measured in corpora. Combining word embeddings into phrasal and sentence vectors can be achieved through various means, often through task-specific models with many parameters of their own, optimised by gradient descent.

In this paper we use unitary matrices in place of arbitrary vector embeddings. Arjovsky et al. (2016) propose *Unitary-Evolution Recurrent Neural Networks* (URNs), to eliminate exploding or vanishing gradients in gradient descent. By the definition of unitary-evolution, at each step, a unitary transformation is applied to the state of the RNN. This means that each input symbol is interpreted as a unitary transformation, or equivalently as a unitary matrix. No activation functions are applied between the time-steps. This design provides a lightweight DNN, with several attractive mathematical and computational properties. URNs are strictly compositional. The effect of embeddings can be analysed independently of context. Therefore the model is transparent, in the sense that it can be analysed by direct inspection, rather than through black box testing methods. So, for example, researchers are forced to resort to probe techniques (Hewitt and Manning, 2019) to ascertain the syntactic structure which transformers and other DNNs represent.

Because of the reversibility of unitary transformations, long distance dependency relations can, in principle, be reliably and efficiently recognised, without additional special-purpose machinery of the kind required in an LSTM. This has been demonstrated to hold for copying and adding tasks (Arjovsky et al., 2016; Jing et al., 2017; Vorontsov et al., 2017) (See also section 6.4).

Here we view the unitary matrices learned by a URN as *word embeddings*. Doing so gives a richer structure to embeddings, with computational and formal advantages that are absent from the traditional vector format that dominates current work in deep learning.

We demonstrate these advantages by applying the URN architecture to two tasks: (i) bracket matching in a generalised Dyck language, and (ii) the more challenging task of subject-verb number agreement in English. These experiments confirm the long-distance capabilities of URNs, even on a linguistically interesting and difficult task.

The richer structure of unitary embeddings permits us to measure the relative effects and distances of different words and phrases. We illustrate the application of such metrics for both experiments.

In section 2 we describe the design of the URN, and our implementation of it. Sections 4 and 5 present our experiments and their results, leverag-

ing the theory presented in section 3.[1] We discuss related work in section 6, and we draw conclusions and sketch future work in section 7.

The computational perspicuity of URNs allows them to be compared to psychologically and neurologically attested models of human learning and representation. Most deep neural networks, particularly powerful transformers, use non-linear activation functions which render their operation opaque and difficult to understand. By contrast, the computations of an URN are explicitly given as simple matrix multiplications, and they are open to inspection at each point in the processing sequence.

## 2 Models

In its full generality, a recurrent network is a function from an input state vector $s_0$ and a sequence of input vectors $x_i$, such that the state at each time-step is a function of the state at the previous step and the input at that step: $s_{i+1} = f(x_i, s_i)$. The function $f$ is constant across steps, and it is called a "cell" of the network.

Since the simple recurrent networks of Elman (1990), the dominant architectures of RNNs, including the influential LSTM (Hochreiter and Schmidhuber, 1997), use non-linear activation functions ($sigmoid$, $tanh$, ReLU) at each time-step. Transformer models, like BERT, are even more opaque in their operations, due the their reliance on a large number of attention heads that apply non-linear functions at each level. By contrast our URNs invoke only linear cells. In fact, the cell that we use is a linear transformation of the unitary space,[2] so that it takes unit state vectors to unit state vectors, hence the term "unitary-evolution". Expressed as an equation, we have $f(x, s) = Q(x)s$, where $Q(x)$ is unitary. Therefore, only state vectors $s_i$ of norm 1 play a role in URNs.

In our implementation of the URN architecture we limit ourselves to real numbers, and so $Q(x)$ is properly described as an orthogonal matrix. We follow this terminology in what follows.

Let $n$ be the dimension of the state vectors $s_i$, and $N$ the length of the sequence of inputs. We will consider only the case of $n$ even. In all our experiments, we take $s_0$ to be the vector $[1, 0, \ldots]$ without loss of generality. For predictions, we extract a probability distribution from state vectors by applying a dense layer with softmax activation to each $s_i$.

We need to ensure that $Q(x)$ is (and remains) orthogonal when it is subjected to gradient descent. In general, subtracting a gradient to an orthogonal matrix does not preserve orthogonality of the matrix. So we cannot make $Q(x)$ a simple lookup table from symbol to orthogonal matrix without additional restrictions. While one could project the matrix onto an orthogonal space (Wisdom et al., 2016; Kiani et al., 2022), our solution is to use a lookup table mapping each word to a skew-hermitian matrix $S(x)$.[3] We follow Hyland and Rätsch (2017) in doing this. We then let $Q(x) = e^{S(x)}$, which ensures the orthogonality of $Q(x)$. It is not difficult to ensure that $S(x)$ is skew-symmetric. It suffices to store only the elements of $S(x)$ above the diagonal, and let those below it be their anti-symmetric image, while the diagonal is set at zero.

Another important issue is that the number of parameters in $S(x)$ grows with the square of $n$. This would entail that doubling a model's power requires quadrupling the number of its parameters. To remedy this problem we limit ourselves to matrices $S(x)$ which have non-zero entries only on the first $k$ rows (and consequently $k$ columns). In this way we limit the total size of the embedding to $(n-1) + (n-2) + \cdots + (n-k+1)$, due to the constraint of symmetry. Consequently, $S(x)$ has at most rank $2k$. Below, we refer to this setup as consisting of *truncated* embeddings.

As an example, the 3×3 skew-symmetric matrix $\begin{pmatrix} 0 & a & b \\ -a & 0 & c \\ -b & -c & 0 \end{pmatrix}$ is 1-truncated if $c = 0$. This truncation reduces its informational content to the single row (and column) $(a \; b)$.

We use the acronym URN to refer to the general class of unitary-evolution networks, $k$-TURN to refer to our specific model architecture with $k$-truncation of embeddings (fig. 1), and Full-URN for our model architecture with no truncation.

We employ a standard training regime for our experiments. We apply a dropout function on both inputs of $f$, so that some entries of $s_i$ or $Q(x_i)$ will be zeroed out according to a Bernoulli distribution

---

[1]The code and relevant linear algebra proofs for our model is available at `https://github.com/GU-CLASP/unitary-recurrent-network`.

[2]The subspace of vectors of unit norm

[3]A matrix $S$ is skew-symmetric iff $S^T = -S$. Here, we rely on the the property that the exponential of any skew-symmetric matrix is orthogonal . The mathematical tools that we employ are standard (Gantmacher, 1959). The key results and their proofs are available at `https://github.com/GU-CLASP/unitary-recurrent-network/blob/main/proofs.pdf`.

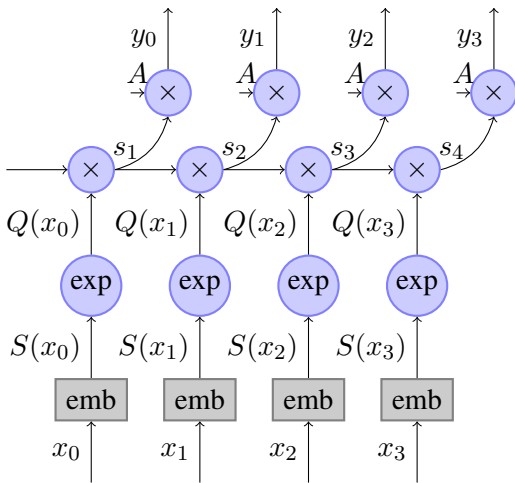

Figure 1: TURN architecture. Each input symbol $x_i$ indexes an embedding layer, yielding a skew-symmetric matrix $S(x_i)$. Taking its exponential yields an orthogonal matrix $Q(x_i)$. Multiplying the state $s_i$ by $Q(x_i)$ yields the next state, $s_{i+1}$.

of rate $\rho$.[4] The embeddings are optimised by means of the Adam gradient descent algorithm (Kingma and Ba, 2014), with no further adjustment. Our implementation uses the TensorFlow (Abadi et al., 2016) framework (version 2.2), including its implementation of matrix exponential.

## 3   Properties of Orthogonal Embeddings

The absence of activation functions in the URN make it more amenable to theoretical analysis than the general class of RNNs with activation functions, including LSTMs and GRUs. The key feature of this design is that the behaviour of the cell is entirely defined by the matrix $Q(x)$, the orthogonal embedding of $x$. The cell only multiplies by word embeddings, and we can focus solely on those embeddings to understand the model.

Since the work of Mikolov et al. (2013), vector embeddings have proven to be an extremely successful modelling tool. However, their structure is opaque. The only way of analysing their relations is through geometric distance metrics like cosine similarity. The unit vectors $u$ and $v$ are deemed similar if $\langle u, v \rangle$ is close to 1. Here we work with orthogonal matrix embeddings, which exhibit much richer structure. We use mathematical analysis to get a better sense of this structure, and relate it to vector embeddings.

**Composition of Embeddings**   A decisive benefit of unitary (and orthogonal) matrix embeddings is that they form a group. We can obtain the inverse of a word embedding simply by transposing it: $Q(x)^{-1} = Q(x)^T$. We can also compose two embeddings to obtain an embedding for the composition. Thanks to the associativity of multiplication, we have $f(x_1, f(x_0, s_0)) = Q(x_1)(Q(x_0)s_0) = (Q(x_1) \times Q(x_0))s_0$. So, we can define the embedding of any sequence as $Q(x_0 \ldots x_i) = Q(x_i) \times Q(x_{i-1}) \times \cdots \times Q(x_0)$. Using this notation, the final state of an URN is $Q(x_0 \ldots x_{N-1})s_0$. Hence, the URN is *compositional by design.*[5]

It is important to recognise that compositionality is strictly a consequence of the structure of a URN. It follows directly from the use of unitary matrix multiplication, through which the successive states in the RNN's processing sequence are computed, without activation functions, It is not necessary to demonstrate this result experimentally, since it is a formal consequence of the associativity of orthogonal matrix multiplication, as shown above. Because URNs do not incorporate additional nonlinear activation functions, a simple matrix is always sufficient to express any combination of word and phrasal embeddings.

**Distance and Similarity**   For vector embeddings, one often uses cosine similarity as a metric of proximity. With unit vectors, this cosine similarity is equal to the inner product $\langle u, v \rangle = \sum_i u_i v_i$. In unitary space, it is equivalent to working with euclidean distance squared, because $\|u - v\|^2 = 2(1 - \langle u, v \rangle)$.

Notions of vector similarity and distance can be naturally extended to matrices. The Frobenius inner product $\langle P, Q \rangle = \Sigma_{ij} P_{ij} Q_{ij}$ extends cosine similarity, and the Frobenius norm $\|A\|^2 = \sum_{ij} A_{ij}^2$ extends euclidean norm. Furthermore, for orthogonal matrices they relate in an analogous way to unit vectors: $\|P - Q\|^2 = 2(n - \langle P, Q \rangle)$.

Why is the Frobenius norm a natural extension of cosine similarity for vectors? It is not merely due to the similarity of the respective formulas.

---

[4]Even though we follow this regime to be standard, experiments indicate that dropout rates appear not critical when we restrict transformations to be unitary.

[5]One might expect that the composition of embeddings can be done at the level of skew-symmetric embeddings: $S(x_0 x_1) = S(x_0) + S(x_1)$. However, this will not work. The law $e^{S_0 + S_1} = e^{S_0} e^{S_1}$ holds only when $S_0$ and $S_1$ commute, which is, in general, not true in our setup. This non-commutativity makes it possible to obtain, by composition, embeddings of higher rank, by which way we make use of all the dimensions of the orthogonal group.

The connection is deeper. A crucial property of the Frobenius inner product (and associated norm) is that it measures the average behaviour of orthogonal matrices on state vectors. More precisely, the following holds: $\mathbb{E}_s[\langle Ps, Qs \rangle] = \frac{1}{n}\langle P, Q \rangle$ , and $\mathbb{E}_s[\|Ps - Qs\|^2] = \frac{1}{n}\|P - Q\|^2$. In sum, as a fallback, one can analyse unitary embeddings using the methods developed for plain vector embeddings. Doing so is theoretically sound. Together with the fact that matrix embeddings can be composed, it means that one can analyse the distances between *phrases*.

**Average Effect**   A useful metric for unitary embeddings is the squared distance to the identity matrix, $\|Q - I\|^2$. By the above result, it is the average squared distance between $s$ and $Qs$ — essentially, the average effect that $Q$ has, relative to the task for which the URN is trained. Note that this sort of metric is unavailable when using opaque vector embeddings. In particular, the norm of a vector embedding is not directly interpretable as a measure of its effect. In the case of an LSTM, for example, vector embeddings first undergo linear transformations *followed by activation functions*, before effecting the state, in several separate stages.

**Signature of Embeddings**   While the average effect is a useful measure, it is rather crude. Averaging over random state vectors considers all features as equivalent. But we might be interested in the effect of $Q$ along specific dimensions, measured separately.

For this purpose, it is useful to note that any orthogonal matrix $Q$ can be decomposed as the effect of $n/2$ independent rotations, in $n/2$ orthogonal planes. The angles of these rotations define how strongly $Q$ effects the state vectors lying in this plane. We refer to such a list of angles as the *signature* of $Q$, and we denote it as $\mathrm{sig}(Q)$. When displaying a signature, we omit any zero angle. This is useful because a $k$-truncated embedding has at most $k$ non-zero angles in its signature. Nonzero angles will be represented graphically as a dial, with small angles pointing up ⊙, and large angles pointing down ⊙.

## 4   Natural Language Agreement Task

It may seem that the extreme simplicity of the TURN architecture renders it unsuitable for any non-trivial processing task. In fact, this is not at all the case.

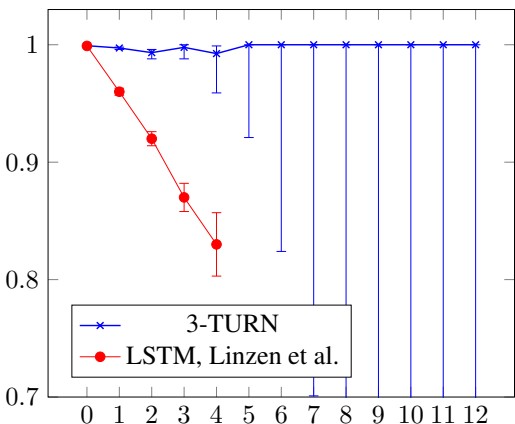

Figure 2: Accuracy per number of attractors for the verb number agreement task. Linzen et al. (2016) do not report performance of their LSTM past 4 attractors. Error bars represent binomial 95% confidence intervals.

Our first experiment applies a TURN to a natural language agreement task proposed by Linzen et al. (2016). This task is to predict the number of third person verbs in English text, with supervised training. In the phrase "The **keys** to the cabinet **are** on the table", the RNN is trained to predict the plural "**are**" rather than the singular "**is**".

The training data is composed of 1.7 million sentences with a selected subject-verb pair, extracted from Wikipedia. The vocabulary size is 50,000, and out-of-vocabulary tokens are replaced by their part-of-speech tag. Training is performed for ten epochs, with a learning rate of 0.01, and a dropout rate of $\rho = 0.05$. We use 90% of the data for training and 10% for validation and testing. A development subset is not necessary since no effort was made to tune hyperparameters. Our first experiment proved sufficient to illustrate our main claims. In any case, a TURN has few hyperparameters to optimise.

Linzen et al. (2016) point out that solving the agreement task requires knowledge of hierarchical syntactic structure. That is, if an RNN captures the long-distance dependencies involved in agreement relations, it cannot rely solely on the linear sequence of nouns (in particular their number inflections) preceding the predicted verb in a sentence. In particular, the accuracy must be sustained as the number *attractors* increases. An attractor is defined as a noun occurring between the subject and the verb which additionally exhibits the wrong number feature required to control the verb. In the above example sentence, "cabinet" is an attractor.

Figure 2 shows the results for a 50-unit TURN

| word | effect | word | effect | word | effect |
|------|--------|------|--------|-------|--------|
| .    | 0.22   | an   | 3.70   | for   | 4.62   |
| the  | 1.44   | as   | 3.76   | in    | 4.62   |
| his  | 1.47   | he   | 3.95   | have  | 4.62   |
| its  | 2.17   | had  | 3.95   | who   | 4.68   |
| also | 2.27   | to   | 3.96   | were  | 4.88   |
| their| 2.54   | a    | 4.06   | that  | 5.00   |
| not  | 2.73   | of   | 4.09   | was   | 5.55   |
| been | 2.82   | from | 4.09   | (     | 5.68   |
| at   | 3.40   | i    | 4.11   | )     | 5.74   |
| or   | 3.46   | it   | 4.14   | are   | 6.25   |
| by   | 3.50   | and  | 4.18   | but   | 6.27   |
| one  | 3.54   | on   | 4.33   | is    | 6.38   |
| this | 3.62   | with | 4.36   | which | 7.75   |
| be   | 3.65   | has  | 4.41   | ,     | 8.35   |

Table 1: Table of average effects for agreement experiment for the most frequent tokens in the corpus, ordered by average effect, from least to greatest

with 3-truncated embeddings for the agreement task, for up to 12 attractors. We see that the TURN "solves" this task, with error rates well under one percent. Crucially, there is no evidence of accuracy dropping as the number of attractors increases. Even though the statistical uncertainty increases with the number of attractors, due to decreasing numbers of examples, the TURN makes no mistakes for the higher number of attractor cases.

### 4.1 Average effect

In this section we illustrate the notion of average effect developed in 3, for this task.

We report the average effect for the embeddings of the most common words in the dataset (table 1), and other selected words and phrases obtained by composition. We stress that this is **not** done by measuring the average effect on the data set; but rather using the formula $\|Q - I\|^2$ for each unitary embedding $Q$. Looking at the table of effects for these words and phrases (ordered from smallest to largest effect) confirms the analysis of 3: tokens which are relevant to the task (e.g. verbs, relative pronouns) generally have a larger effect than those which are not (e.g. the dot, "not").

We also computed the distance between pairs of the most frequent nouns, with both singular and plural inflections (table 2). We observe, as our account predicts, that nouns with the same number inflection tend to be grouped (with a distance of 7.5 or less between them), while nouns with differing numbers are further apart (with a distance of 7.5 or

more).

## 5 Dyck-language modelling task

To evaluate the *theoretical* long-distance modelling capabilities of an RNN in a way that abstracts away from the noise in natural language, one can construct synthetic data. Following Bernardy (2018) we use a (generalised) Dyck language. This language is composed solely of matching parenthesis pairs. So the strings "{([])}<>" and "{()[<>]}" are part of the language, while "[}" is not. This experiment is an idealised version of the agreement task, where opening parentheses correspond to subjects, and closing parentheses to verbs. An attractor is an opening parenthesis occurring between the pair, but of a different kind. Matching of parentheses corresponds to agreement. Because we use five distinct kinds of parentheses, the majority class baseline is at 20%. This makes it easier to evaluate the performance of a model on the matching task than for the third person agreement task, where the majority class baseline for the training corpus is above 70%.

We complicate the matching task with an additional difficulty. We vary the nesting depth between training and test phases. The *depth* of the string is the maximum nesting level reached within it. For instance "[{}]" has depth 2, while "{([()]<>)}" has depth 4. In this task, we use strings with a length of exactly 20 characters. We train on 102,400 randomly generated strings, with maximum depth 3, and test it on 5120 random strings of maximum depth 10. Training is performed with a learning rate of 0.01, and a dropout rate of $\rho = 0.05$, for 100 epochs.

The training phase treats the URN as a generative language model, applying a cross-entropy loss function at each position in the string. At test time, we evaluate the model's ability to predict the right kind of closing parenthesis at each point (this is the equivalent of predicting the number of a verb). We ignore predictions regarding opening parentheses, because they are always acceptable for the language.

We ran three versions of this experiment. One with truncated embeddings, one with full embeddings, and a third using a baseline RNN with full embeddings that are not constrained to be orthogonal. In all cases, the size of matrices is 50 by 50. We report accuracy on the task by number of attractors in fig. 3.

| | article | year | area | world | family | articles | years | areas | worlds | families |
|---|---|---|---|---|---|---|---|---|---|---|
| article | 0.00 | 7.04 | 6.51 | 6.89 | 5.82 | 9.26 | 9.84 | 10.01 | 10.87 | 9.39 |
| year | 7.04 | 0.00 | 7.62 | 6.30 | 5.38 | 8.22 | 9.06 | 9.75 | 10.14 | 8.64 |
| area | 6.51 | 7.62 | 0.00 | 6.42 | 6.34 | 9.57 | 9.70 | 10.39 | 11.63 | 10.39 |
| world | 6.89 | 6.30 | 6.42 | 0.00 | 5.17 | 7.32 | 8.82 | 9.17 | 9.13 | 7.83 |
| family | 5.82 | 5.38 | 6.34 | 5.17 | 0.00 | 7.71 | 7.72 | 8.78 | 9.49 | 8.82 |
| articles | 9.26 | 8.22 | 9.57 | 7.32 | 7.71 | 0.00 | 5.11 | 4.79 | 4.28 | 4.57 |
| years | 9.84 | 9.06 | 9.70 | 8.82 | 7.72 | 5.11 | 0.00 | 6.42 | 6.61 | 7.14 |
| areas | 10.01 | 9.75 | 10.39 | 9.17 | 8.78 | 4.79 | 6.42 | 0.00 | 5.93 | 6.09 |
| worlds | 10.87 | 10.14 | 11.63 | 9.13 | 9.49 | 4.28 | 6.61 | 5.93 | 0.00 | 7.79 |
| families | 9.39 | 8.64 | 10.39 | 7.83 | 8.82 | 4.57 | 7.14 | 6.09 | 7.79 | 0.00 |

Table 2: Distances between embeddings of most frequent nouns and their plural variants. Words which can be both nouns and verbs were excluded.

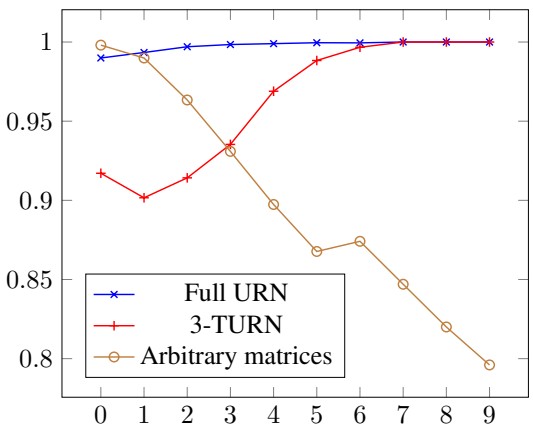

Figure 3: Accuracy of closing parenthesis prediction by number of attractors.

| | ⊘ | ⊖ | ⊘ |
|---|---|---|---|
| ⊘ | 0.33 | 0.35 | 1.35 |
| ⊖ | 0.46 | 1.73 | 0.2 |
| ⊘ | 1.09 | 0.2 | 0.34 |

Table 3: Similarity for each pair of rotation planes, for the embeddings of ( and [. Headers show the rotation effected on the compared planes. A value of 2 indicates that the planes are equal (up to rotation of the basis vectors), and a value of 0 indicates that they are orthogonal.

We note that even the baseline model is capable of generalising to longer distances. Up to 9 attractors, it achieves performance that is well above a majority class baseline (20%). However, it shows steadily decreasing accuracy as the number of attractors increases.

By contrast, the URN models remain accurate as the number of attractors grows. Perhaps surprisingly, the URN improves in relation to the number of attractors. We will solve this apparent puzzle below, through analysis of the embeddings. The explanation will hinge on the fact that truncating embeddings affects performance only when the number of attractors is low.

Comparing the arbitrary embeddings model with with full URN highlights the importance of limiting the network to orthogonal matrices. The performance of the full URN is better over the long term and in general, with a validation loss of 1.47213 compared to 1.52914 for the arbitrary case. This happens despite the fact that the orthogonal system

is a special case of the arbitrary network, and so orthogonal embeddings are, in principle, available to the baseline RNN. But it is not able to converge on the preferred solution (even for absolute loss). In sum, restricting to orthogonal matrices acts like a regularising constraint which offers a significant net benefit in generalisation and tracking power.

## 5.1 Analysis

As in the previous experiment, matrix embeddings can be analysed regardless of contexts, offering a direct view of how the model works. We consider the embeddings produced by training the 3-TURN model, and we start with the embeddings of individual characters and their signatures (table 4). The average effect, and even the signatures of all embeddings are strikingly similar. This does not imply that they are *equal*. Indeed, they rotate different planes.

We see in table 3 that the planes which undergo rotation by similar angles are far from orthogonal to each other— one pair even exhibits a similarity of 1.73. This corresponds to the fact that the transformations of ( and [ manipulate a common subset of coordinates. On the other hand, those planes that undergo rotation by different angles tend to be in a

| character | average effect | signature |
|---|---|---|
| ( | 14.79 | ◐⊘◑ |
| < | 14.34 | ◐⊘◑ |
| { | 13.98 | ◐⊘◑ |
| [ | 14.25 | ◐⊘◑ |
| + | 14.20 | ◐⊘◑ |
| ) | 14.85 | ◐⊘◑ |
| > | 14.42 | ◐⊘◑ |
| } | 14.07 | ◐⊘◑ |
| ] | 14.34 | ◐⊘◑ |
| − | 14.26 | ◐⊘◑ |
| ( ) | 0.06 | ◐◐◐◐ |
| < > | 0.06 | ◐◐◐◐ |
| { } | 0.07 | ◐◐◐◐ |
| [ ] | 0.06 | ◐◐◐◐ |
| +− | 0.06 | ◐◐◐◐ |

Table 4: Average effect and signatures of parenthesis embeddings and matching pairs.

closer to orthogonal relationship.

**Composition of Matching Parentheses**  To further clarify the formal properties of our model let's look at the embeddings of matching pairs, computed as the product of the respective embeddings of the pairs. Such compositions are close to identity (table 4). This observation explains the extraordinarily accurate long-distance performance of the URN on the matching task. Because a matching pair has essentially no effect on the state, by the time all parentheses have been closed, the state returns to its original condition. Accordingly, the model experiences the highest level of confusion when it is *inside* a deeply nested structure, and *not* when a deep structure is inserted between the governing opening parenthesis and the prediction conditioned on that parenthesis.

# 6   Related Work

## 6.1   Explainable NLP

It has frequently been observed that DNNs are complex and opaque in the way in which they operate. It is often unclear how they arrive at their results, or why they identify the patterns that they extract from training data. This has given rise to a concerted effort to render deep learning systems explainable (Linzen et al., 2018, 2019). This problem has become more acute with the rapid development of very large pre-trained transformer models (Vaswani et al., 2017), like BERT (Devlin et al., 2018), GPT2

(Solaiman et al., 2019), GPT3 (Brown et al., 2020), and XLNet (Yang et al., 2019).

URNs avoid this difficulty by being compositional by design. If they prove robust for a wide variety of NLP tasks, they will go some way to solving the problem of explainability in deep learning.

**Learning Agreement**  The question of whether generative language models can learn long-distance agreement was proposed by Linzen et al. (2016). If accuracy is insensitive to the number of attractors, then we know that the model can work on long distances. The results of Linzen et al. (2016) are inconclusive on this question. Even though the model does better than the majority class baseline for up to four attractors, accuracy declines steadily as the number of attractor increases. This trend is confirmed by Bernardy and Lappin (2017), who ran the same experiment on a larger dataset and thoroughly explored the space of hyperparameters. It is also confirmed by Gulordava et al. (2018), who analysed languages other than English. Marvin and Linzen (2018) focused on other linguistic phenomena, reaching similar conclusions. Lakretz et al. (2021) recently showed that an LSTM may extract bounded nested tree structures, without learning a systematic recursive rule. These results do not hold directly for BERT-style models, because they are not generative, even though Goldberg (2019) provides a tentative approach. For a more detailed review of these results, see the recent account of Lappin (2021).

Our experiment shows that URNs can surpass state of the art results for this kind of task. This is not surprising. URNs are designed so that they *cannot forget information*, and so it is expected that they will perform well on tracking long distance relations. The conservation of information is explained by the fact that multiplying by an orthogonal matrix conserves cosine similarities: $\langle Qs_0, Qs_1 \rangle = \langle s_0, s_1 \rangle$. Therefore any embedding $Q$, be it of a single word or of a long phrase, maps a change in its input state to an equal change in its output state. Considering all possible states as a distribution, $Q$ conserves the density of states. Hence, contrary to the claims of Sennhauser and Berwick (2018), URNs demonstrate that a class of RNNs can achieve rule-like accuracy in syntactic learning.

**Dyck Languages**  Elman (1991) already ob-

served that it is useful to experiment with artificial systems to filter out the noise of real world natural language data. However, to ensure that the model actually learns recursive patterns instead of bounded-level ones, it is necessary to test on more deeply nested structures than the ones that the model is trained on, as we did. Generalised Dyck languages are ideal for this purpose (Bernardy, 2018). While LSTMs (and GRUs) exhibit a certain capacity to generalise to deeper nesting their performance declines in proportion to the depth of the nesting, as is the case with their handling of natural language agreement data. Other experimental work has also illustrated this effect (Hewitt et al., 2020; Sennhauser and Berwick, 2018). Similar conclusions are observed for generative self-attention architectures (Yu et al., 2019), while BERT-like, non-generative self-attention architectures simply fail at this task (Bernardy et al., 2021).

By contrast URNs achieve excellent performance on this task, without declining in relation to either depth of nesting or the number of attractors. Careful analysis of the learned embeddings explains this level of accuracy in a principled way, as the direct consequence of their formal processing design.

## 6.2 Quantum-Inspired Systems

Unitary matrices are essential elements of quantum mechanics, and quantum computing. There, too, they insure that the relevant system does not lose information through time.

Coecke et al. (2010); Grefenstette et al. (2011) propose what they describe as a quantum inspired model of linguistic representation. It computes vector values for sentences in a category theoretic representation of the types of a pregroup grammar (Lambek, 2008). The category theoretic structure in which this grammar is formulated is isomorphic with the one for quantum logic.[6]

A difficulty of this approach is that it requires the input to be already annotated as parsed data. Another problem is that the size of the tensors associated with higher-types is very large, making them hard to learn. By contrast, URNs do not require a syntactic type system. In fact, our experiments indicate that, with the right processing network, it is possible to learn syntactic structure and semantic composition from unannotated input.

Compositionality of phrase and sentence matri-

---

[6]See Lappin (2021) for additional discussion of this theory.

ces is intrinsic to the formal specification of the network.

## 6.3 Tensor Recurrent Neural Networks

Sutskever et al. (2011) describe what they call a "tensor recurrent neural network" in which the transition matrix is determined by each input symbol. This design appears to be similar to URNs. However, unlike URNs, they use non-linear activation functions, and so they inherit the complications that these functions bring.

## 6.4 Unitary-Evolution Recurrent Networks

Arjovsky et al. (2016) proposed Unitary-Evolution recurrent networks to solve the problem of exploding and vanishing gradients, caused by the presence of non-linear activation functions. Despite this, Arjovsky et al. (2016) suggest that they use ReLU activation between time-steps, unlike URNs. Moreover, we are primarily concerned with the structure of the underlying unitary embeddings. The connection between the two lines work is that, if an RNN suffers exploding/vanisihing gradients, it cannot track long-term dependencies.

Arjovsky et al. (2016)'s embeddings are computationally cheaper than ours, because they can be multiplied in linear time. Like us, they do not cover the whole space of unitary matrices. Jing et al. (2017) propose another representation which is computationally less expensive than ours, but which has asymptotically the same number of parameters. A third option is let back-propagation update the unitary matrices arbitrarily $n \times n$, and project them onto the unitary space periodically (Wisdom et al., 2016; Kiani et al., 2022).

Because we use a fully general matrix exponential implementation, our model is computationally more expensive than all the other options mentioned above. We can however report that when experimenting with the unitary matrix encodings Jing et al. (2017) and Arjovsky et al. (2016), we got much worse results for our experiments. This may be because we do not include a ReLU activation, while they do use one.

To the best of our knowledge, no previous study of URNs has addressed agreement or other language modelling tasks. Rather, they have been directed at data-copying tasks, which is of limited linguistic interest. This includes the work of Vorontsov et al. (2017), even though it is ostensibly concerned with long distance dependencies.

## 7  Conclusions and Future work

In conclusion, we have shown that the URN is a useful architecture for syntactic tasks, for which it can reach or surpass state-of-the art precision. We strongly suspect that it will also prove effective for NLP tasks requiring fine-grained semantic knowledge. Unlike other DNNs, a URN is transparent and mathematically grounded in straightforward operations of linear algebra. It is possible to trace and understand what is happening at each level of the network, and at each point in the sequence that makes up the processing flow of the network.

Additionally, URNs learn *unitary embeddings*. These offer two important advantages. First, they have a rich internal structure from which we can analyse the learned model. Second they handle compositionality without stipulated constraints, or additional mechanisms. Therefore we can obtain unitary embeddings for any phrase or sentence.

The refined distance, effect, and relatedness metrics that unitary embeddings facilitate, open up the possibility of more interesting procedures for identifying natural syntactic and semantic word classes. These can be textured and dynamic, rather than static. They can focus on specific dimensions of meaning and structure, and they can be driven by specific NLP tasks. If additional types of input data are encoded in a matrix, such as visual content, then these classes could also be grounded in extralinguistic contexts.

In order to render URNs efficient, it is necessary to reduce the number of parameters from which the matrix can be derived. We found that a simple $k$-truncation of underlying anti-symmetric matrices is a useful strategy to limit the size of word embeddings. It also makes the learned embeddings more accessible to formal analysis, because they can be decomposed as rotations along $k$ planes. For the tasks that we considered, truncation does not seriously degrade the performance of the TURN model. Kiani et al. (2022) recently applied this strategy to another subset of tasks, suggesting general viability of this strategy.

In preliminary work we have applied URNs to the recognition of mildly context-sensitive languages containing cross serial dependencies of the sort found in Swiss German and in Dutch. The performance of the model is even more robust and stable than it is for the agreement tasks reported here. We will be extending this work to a variety of other linguistically and cognitively interesting NLP tasks.

Given the radical computational transparency of URN architecture, these models are natural candidates for comparison with human processing systems, both at the neurological level, and on more abstract psychological planes. Identifying and measuring the content of their acquired knowledge for particular tasks can be done through direct observation of their processing patterns, and the application of straightforward distance metrics. In this respect they are of particular interest in the study of the cognitive foundations of linguistic learning and representation.

## 8  Acknowledgements

The research reported in this paper was supported by grant 2014-39 from the Swedish Research Council, which funds the Centre for Linguistic Theory and Studies in Probability (CLASP) in the Department of Philosophy, Linguistics, and Theory of Science at the University of Gothenburg We thank three anonymous reviewers for their helpful comments on an earlier draft of this paper. We presented the main ideas of this paper to the CLASP Seminar, in December 2021, and to the Cognitive Science Seminar of the School of Electronic Engineering and Computer Science, Queen Mary University of London, in February 2022. We are grateful to the audiences of these two events for useful discussion and feedback.

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
