# OpenReview forum: "A Neural Model for Compositional Word Embeddings and Sentence Processing"
_aclweb.org/ACL/2022/Workshop/CMCL — CMCL 2022_

### Official Review · Reviewer_7SS9 · 2022-03-21
**An interesting paper which proposes a new neural embedding network which emphasizes compositionality and sentence embedding. The network is linear and hence arguably paves the way to interpretable comparisons with humans.**

**Rating:** 6
**Confidence:** 3

**Review:**

This paper proposes a new neural model for word embeddings, which uses unitary matrices as the primary device for encoding lexical information. This model does not employ non-linear activation functions, and can hence be used with linear algebra tools. The model emphasizes compositionality. The model is interesting but I am not sure the work is directly related to the theme of the workshop (which is the main reason that I did not give the work a higher score).

---

### Official Review · Reviewer_PSXn · 2022-03-24
**Robust mathematical implementation of a new RNN which provide a totally explainable deep learning model for natural language learning and representation**

**Rating:** 7
**Confidence:** 3

**Review:**

This paper introduces a new Recurrent Neural Network that includes Unitary Matrices at each step. The cell uses a linear transformation of the space instead of applying non-linear activations.
The authors evaluate how the RNN captures long-distance dependencies on two tasks: number agreement and Dick-language modeling (a symbolic synthetic language). The tasks are well-described and coherent with their research questions; results demonstrate how this mathematical implementation can help understand the linguistic representations created by deep neural architectures.

The paper offers a solid mathematical explanation and motivation for the implementation. The authors make everything as precise as possible, even if mathematical sections are a bit demanding for non-mathematicians. The only point that maybe could be more transparent regards the "average effect." In section 4.1, the exposition is condensed into very few lines. I suggest providing more details about this measure, making it more painless for a larger audience.

The only limitation of the presented work is perhaps that one must train the network for every specific task in a supervised way.

The overall judgment is favorable. I would be glad to see this model tested on semantic compositional tasks in NLP.

---

### Official Review · Reviewer_AkA5 · 2022-03-28
**An interesting new model of language representation but not entirely convincing**

**Rating:** 5
**Confidence:** 4

**Review:**

This paper presents a new model for word embeddings that is based on the Unitary-Evolution Recurrent Neural Network architecture. The main advantage of the proposed embeddings is that they don't rely on non-linear cell operations making the resulting representations more interpretable and natively compositional.

While the promise of a more interpretable language representation model is very interesting, I'm not entirely convinced the new embeddings provide more intuitive explanations of what is being represented. The analysis still relies on linear algebra operations and interpretation of arbitrary distance values. The examples of syntactic and semantic closeness are equally transparent in the geometric interpretation of traditional word vectors and the effect size is easily mapped on the attention weight of regular attention-based DNN. In addition, we didn't see a concrete example of the advantage of URNs being able to handle compositionality natively. Ideally, there would be a "head-to-head" comparison (either qualitative or quantitative) between the proposed unitary embeddings and the more established techniques.

Regarding the model's performance on a language task, despite the claims of the authors, there is no comparison with state-of-the-art methods for agreement task. For example, Goldberg (2019) achieves very similar results (up to 4 attractors) to the ones reported for TURN.

There are mentions of the current model's higher computational cost, but there are no figures to demonstrate whether this type of modeling can be practically used instead of the current generation of Transformer-based architectures.

Finally, despite claims to this effect, there is no evidence of the proposed model's neurological validity. The reference to models of human learning and representation in the Introduction (lines 87-89) is vague and lacks supporting citations.

Overall, while the use of URNs as embeddings is an interesting concept to explore, I'm not entirely sure this paper has created a convincing account.


## Comments to authors
I didn't find the mathematical (modeling) sections of the paper very easy to follow. I would highly recommend the use of a running example throughout sections 2 and 3.

To prove the claim that "[unitary embeddings] can focus on specific dimensions of meaning and structure, and they can be driven by specific NLP tasks" it would be very interesting to show the same words being characterised differently due to the task (e.g. showing that the distances in Table 2 flip to group together semantically similar concepts rather than syntacticly similar ones).

---

### Decision · Program_Chairs · 2022-03-29

Accept